# Protective Immunity Induced by Virus-Like Particle Containing Merozoite Surface Protein 9 of *Plasmodium berghei*

**DOI:** 10.3390/vaccines8030428

**Published:** 2020-07-30

**Authors:** Su-Hwa Lee, Hae-Ji Kang, Ki-Back Chu, Swarnendu Basak, Dong-Hun Lee, Eun-Kyung Moon, Fu-Shi Quan

**Affiliations:** 1Department of Biomedical Science, Graduate School, Kyung Hee University, Seoul 02447, Korea; dltnghk228@nate.com (S.-H.L.); haedi1202@naver.com (H.-J.K.); kbchu@khu.ac.kr (K.-B.C.); swarnendubasak.vb@gmail.com (S.B.); hihool@naver.com (D.-H.L.); 2Department of Medical Zoology, Kyung Hee University School of Medicine, Seoul 02447, Korea; ekmoon@khu.ac.kr; 3Department of Medical Research Center for Bioreaction to Reactive Oxygen Species and Biomedical Science Institute, School of Medicine, Graduate School, Kyung Hee University, Seoul 02447, Korea

**Keywords:** *Plasmodium berghei*, merozoite surface protein 9, vaccine, virus-like particle

## Abstract

Merozoite surface protein 9 (MSP-9) from *Plasmodium* has shown promise as a vaccine candidate due to its location and possible role in erythrocyte invasion. In this study, we generated virus-like particles (VLPs) targeting *P. berghei* MSP-9, and investigated the protection against lethal doses of *P. berghei* in a mouse model. We found that VLP vaccination induced a *P. berghei*-specific IgG antibody response in the sera and CD4^+^ and CD8^+^ T cell populations in blood compared to a naïve control group. Upon challenge infection with *P. berghei*, vaccinated mice showed a significant increase in CD4^+^ and CD8^+^ effector memory T cell and memory B cell populations. Importantly, MSP-9 VLP immunization inhibited levels of the pro-inflammatory cytokines IFN-γ and IL-6 in the spleen and parasite replication in blood, resulting in significantly prolonged survival time. These results suggest that the MSP-9 VLP vaccine may constitute an effective malaria vaccine.

## 1. Introduction

Malaria remains a severe public health problem, causing significant economic losses worldwide. The widespread nature of malaria throughout the tropical and the subtropical regions of the world makes it one of the most important parasitic diseases infecting humans. The WHO estimates 300–500 million cases of malaria, with over one million deaths each year [1]. There have been many efforts to develop an effective malaria vaccine. Nonetheless, effective malaria vaccines are currently lacking partially due to its complex life cycle, which presents a greater challenge than most microbes. The most recently developed recombinant vaccine RTS,S/AS01 was engineered by fusing the proteins expressed by repeat and T cell epitope genes in the pre-erythrocytic circumsporozoite protein (CSP) of the *Plasmodium falciparum* malaria parasite with the viral envelope protein of hepatitis B virus (HBsAg), and has undergone Phase 3 clinical trials [2,3]. This recombinant malaria vaccine with an adjuvant (RTS, S/AS01) showed low to moderate efficacy (26–50%) in children with multiple immunizations of four injections [2,3]. Currently manufactured malaria vaccines have low efficacy and alternative vaccine study is urgently needed.

In an infection with *Plasmodium*, merozoite surface protein plays an important role in the invasion of red blood cells, and blood stage invasion is closely related to the occurrence of malaria. An important part of developing malaria vaccines is understanding the merozoite invasion and the host immune response to merozoite antigens in the blood stage of infection [4]. Merozoite surface protein 9 (MSP-9) is one of the antigens located within the merozoite surface and parasitophorous vacuole of the infected erythrocytes [5,6]. MSP-9 was first identified in *P. falciparum* and described by Stahl et al. [5,6] as an acidic basic repeat antigen (ABRA) that responded with human immune sera [6,7]. There is experimental evidence that MSP-9 participates in merozoite release from infected erythrocytes and binds to the human erythrocyte surface via N-terminal cysteine-rich regions or high binding activity peptides (HBAPs) [8,9]. MSP-9 has been suggested to be an important protein in vaccine development due to its role in erythrocyte invasion and immunogenicity in animal models [10,11,12,13].

Virus-like particles (VLPs) contain the main immunological characteristics of viruses, such as repetitive surfaces and particulate structures, and they induce innate immunity through the activation of pathogen-associated molecular pattern (PAMP) recognition receptors [14]. VLPs are easily recognized by the immune system and present clearer forms of antigen than subunit vaccines based on recombinant proteins [15]. In addition, VLPs effectively express robust immune responses in various cells of the humoral and cellular immune system without adjuvants [14,16,17,18,19,20,21,22]. In our previous study, we evaluated the vaccine efficacies in mice immunized with VLPs containing apical membrane antigen 1 (AMA-1) of *Plasmodium berghei* [23]. AMA-1 VLP immunization effectively elicited humoral and cellular immune responses in mice [23]. We sought to investigate in depth the immune response resulting from MSP-9 VLP immunization, using the advantage of the VLP vaccine platform.

*P. berghei*, a rodent malaria parasite, has been extensively used in vaccine development studies [24]. The structure, physiology, and life cycle of *P. berghei* have been documented to be analogous to human and other primate malaria parasites [25]. In the current study, we used BALB/c mice as animal models and the murine malaria parasite *P. berghei* as a challenge infection model. We generated a VLP vaccine containing *P. berghei* MSP-9. Our results clearly demonstrated that VLP vaccine administration elicits a parasite-specific IgG antibody response and humoral and cellular immune responses and significantly reduces the pro-inflammatory cytokines IFN-γ and IL-6 in the spleen and parasitemia in the blood, resulting in a significantly prolonged survival time.

## 2. Materials and Methods

### 2.1. Ethics Statement

All animal experiments and husbandry involved in these studies were conducted under the guidelines of Kyung Hee University IACUC (permit number: KHSASP (SE)-19-188). All animal procedures performed in this study were reviewed, approved, and supervised by Kyung Hee University IACUC. The research staff was trained in animal care and handling. They received the certificate of completion for the Animal Welfare and Ethics Course from CITI. All surgery was performed under isoflurane anesthesia (BSL2). All efforts were made to minimize animal suffering.

### 2.2. Animals, Parasites, Cells, and Antibodies

Six-week-old female BALB/c mice were obtained from NARA Biotech (Seoul, Korea). The *Plasmodium berghei* ANKA strain 2.34 was maintained in mice by serial intraperitoneal passage. *Spodoptera frugiperda* Sf9 cells were used for the production of recombinant baculovirus (rBV) and VLPs. They were maintained in serum-free SF900 II medium (Invitrogen, Carlsbad, CA, USA) in spinner flasks at 27 °C and 130–140 rpm. *P. berghei*-infected sera from mice were collected through retro-orbital plexus puncture. Horseradish peroxidase (HRP)-conjugated goat anti-mouse immunoglobulin G (IgG) was purchased from Southern Biotech (Birmingham, AL, USA). Monoclonal mouse anti-M1 antibody was purchased from Abcam (Cambridge, UK).

### 2.3. Plasmodium Berghei Antigen Preparation

*P. berghei* ANKA strain 2.34 antigen was prepared as described previously [26,27]. The *P. berghei*-infected red blood cells (RBCs) were collected from the whole blood of mice with parasitemia exceeding 20% by low-speed centrifugation (1500 rpm for 10 min) at 4 °C. The RBC pellets were lysed with equal volumes of 0.15% saponin in PBS for 10 min at 37 °C and the released parasites were pelleted and washed three times with PBS (13,000 rpm for 1 min at 4 °C). Parasites were sonicated twice for 30 s at 40 Hz on ice and stored at −20 °C until used. *P. berghei* antigen was used as a coating antigen for ELISA.

### 2.4. Generation of Plasmids, Recombinant Baculovirus, and Virus-Like Particles

The total RNA was extracted from the *Plasmodium berghei* ANKA strain 2.34 using an RNeasy Mini Kit (Qiagen, Valencia, FL, USA). The total RNA was reverse transcribed to cDNA using a Prime Script 1st strand cDNA Synthesis Kit according to the manufacturer’s instructions (Takara, Otsu, Japan). The complete open reading frame (ORF) of merozoite surface protein 9 (MSP-9; GenBank accession no. XM_022858221.1) was amplified by a polymerase chain reaction (PCR) from cDNA using designed specific primers (forward primer: 5′- AAA**GAATTC**ATGAAGATAAGTATCGTGGCA -3′ and reverse primer: 5′- TTA**AAGCTT**TTATGCTTGGTTTGGAGCTGG -3′), with *EcoR*I and *Hind*III restriction enzyme sites, respectively (bold). The PCR product was inserted into the pFastBac vector (Invitrogen, Calsbad, CA, USA). For influenza M1 gene cloning, procedures have been described previously [28]. A construct containing MSP-9 in pFastBac was confirmed by DNA sequencing and transformed into DH10Bac. The bacmid containing the MSP-9 gene was extracted using a FavorPrep Gel Purification Kit (Favorgen, Cheshire, UK) and stored at −20 °C until used. Recombinant baculoviruses expressing MSP-9 or M1 were produced as described previously [29,30]. To produce VLPs expressing MSP-9 and M1, Sf9 cells were co-infected with recombinant baculovirus presenting MSP-9 or influenza M1 as a core protein, as described previously [28,30]. M1 VLPs were prepared by infecting Sf9 cells with rBVs expressing the M1 protein. Sf9 cell culture supernatants were collected on day 3 post infection and centrifuged at 6000 rpm for 30 min at 4 °C to remove cells. VLPs were purified as described previously [29,30]. VLPs were stored at 4 °C until used.

### 2.5. Characterization of VLPs

Purified MSP-9 VLPs were negatively stained on the grid and observed under a transmission electron microscope (TEM) (JEOL 2100, JEOL USA, Inc.; Peabody, MA, USA) [31]. The MSP-9 VLPs were characterized by western blot, as described previously [29]. Briefly, proteins from SDS-PAGE gel were transferred onto PVDF membranes (Merck Millipore, Burlington, MA, USA) and probed with primary antibody (*P. berghei* ANKA anti-polyclonal serum, diluted 1:500 with 2% BSA in TBST), followed by incubation with a secondary antibody (HRP-conjugated anti-mouse IgG, diluted 1:2000 in 5% skimmed milk). The amount of M1 protein was determined with monoclonal mouse anti-M1 antibody (diluted 1:1000 with 2% BSA in TBST).

### 2.6. Mice Immunization and Challenge

BALB/c mice were randomly divided into three experimental groups (naïve, naïve challenge, MSP-9 VLP; *n* = 12 per group) to receive VLP immunization. Mice were intramuscularly immunized with MSP-9 VLPs (100 µg per mice) at weeks 0 and 4. Four weeks after the 2nd immunization, mice were challenged with 1%/100 µL (1 × 10^4^) of *P. berghei* by intraperitoneal (IP) injection, as described previously [23]. Mice from each group were sacrificed at day 6 post challenge. Mice blood and spleen samples were collected. The remaining mice were observed daily to monitor changes in body weight and mice that lost 20% of their body weight were humanely euthanized.

### 2.7. Antibody Responses in Sera

Mice sera were collected from all groups 4 weeks after prime and boost immunization. Sera from naïve mice were used as a negative control. Antibody responses against *P. berghei* Ag, MSP-9 VLPs, and M1 VLPs were determined by enzyme-linked immunosorbent assay (ELISA), as described previously [30]. Briefly, 96-well flat-bottomed immunoplates were coated with 100 µL of *P. berghei* antigen, MSP-9 VLPs, and M1 VLPs at a final concentration of 2, 0.5, and 0.5 µg/mL, respectively, in 0.05 M carbonate bicarbonate buffer (pH 9.6) per well at 4 °C overnight. Then, 100 µL of serially diluted serum samples were loaded into respective wells and incubated for 1 h 30 min at 37 °C, as a primary antibody response. HRP-conjugated goat anti-mouse IgG (100 µL/well, diluted 1:2000 in PBST) was used to determine IgG response. To test whether the MSP-9 protein in VLPs was in a properly folded conformation, the MSP-9 VLPs were tested for antibody recognition using an ELISA as described above. *P. berghei* and *Trichinella spiralis* (TS) antigens were used as positive and negative controls.

### 2.8. Immune Cell Responses by Flow Cytometry

To determine the immune cell responses, the levels of T cell and B cell populations from the blood and splenocytes of mice were investigated by flow cytometry. Blood (150 µL per mouse) was collected and red blood cells were removed using RBC lysis buffer (Sigma-Aldrich, St. Louis, MO, USA). Collected immune cells were counted and 1 × 10^5^ cells in each tube were used for FACS analysis. CD4^+^ and CD8^+^ T cells from the blood were detected after the 1st and 2nd immunization and on day 6 after the challenge infection. The levels of memory T and B cells from mouse spleen were detected on day 6 after the challenge infection. The immune cells (1 × 10^6^ cell in each tube) in staining buffer (2% bovine serum albumin and 0.1% sodium azide in 0.1 M PBS) were incubated at 4 °C for 15 min with Fc Block (clone 2.4G2; BD Biosciences, San Jose, CA, USA). For staining, the cells were incubated with surface antibodies (CD3e-PE-Cy7, CD4-FITC, CD8a-FITC or PE, CD44-PE, CD62L-APC, B220-FITC, IgG1-PE, CD27-PE-Cy7; BD Biosciences, CA, USA) at 4 °C for 30 min. After incubation, the cells were washed with staining buffer two times before acquisition using a BD Accuri C6 Flow Cytometer (BD Biosciences, San Jose, CA, USA). Data were analyzed using C6 Analysis software (BD Biosciences, San Jose, CA, USA).

### 2.9. Inflammatory Cytokine Analysis

Spleen samples were collected at day 6 post challenge to determine the cytokine response to challenge infection. Splenocytes were prepared in RPMI-1640 media (Lonza, Basel, Switzaland). The levels of the cytokines IFN-γ and IL-6 were determined with a DB OptEIA set (BD Biosciences, CA, USA). Cytokine concentrations were quantified by an ELISA reader (λ = 490 nm, *n* = 3) with reference to standard curves constructed with BD IFN-γ or IL-6 stock solution. All procedures were performed according to the DB OptEIA set manufacturer’s instructions.

### 2.10. Parasitemia

For the staining of infected RBCs, 2 µL of blood obtained from the retro-orbital plexus puncture of infected mice was collected into a 1.5 mL tube containing 100 µL of a premixed stock of PBS with 500 U/mL of heparin. RBCs from *P. berghei* (ANKA)-infected mice were stained using 1 µL SYBR Green I (Invitrogen, Carlsbad, CA, USA). The samples were incubated at 37 °C for 30 min in the dark. After incubation, 0.1 M PBS was added to each sample and flow cytometry was performed [32].

### 2.11. Statistics

All parameters were recorded for individuals within all groups and data sets are presented as mean ± SEM. Statistical comparisons of data were carried out by a one-way ANOVA with Tukey’s post hoc test or a Student’s *t*-test using PC-SAS 9.4 (SAS Institute, Cary, NC, USA). A *p*-Value < 0.05 was considered to be significant.

## 3. Results

### 3.1. Generation of MSP-9 Virus-Like Particle Vaccines

The *P. berghei* MSP-9 gene was cloned into a pFastBac vector and the construct was checked using restriction enzymes *EcoR*I/*Hind*III. The restriction of *Plasmodium berghei* MSP-9 in pFastBac expressing vectors was confirmed (Figure 1A). VLPs were generated in Sf9 cells co-infected with recombinant baculoviruses (rBVs) expressing *P. berghei* MSP-9 and influenza matrix protein 1 (M1) following a procedure described previously [30]. As shown in Figure 1B,C, the morphology of the VLPs was examined by electron microscopy and the expression was confirmed by a western blot. MSP-9 VLPs displayed a spherical morphology with a size of approximately 40–120 nm in diameter. They exhibited antigen spikes on their surfaces (Figure 1B). Western blot analysis, using an anti-*P. berghei* polyclonal antibody and an anti-M1 monoclonal antibody, confirmed the presence of *P. berghei* MSP-9 and influenza M1 in VLPs (Figure 1C). As seen in Figure 1D, *P. berghei* antibodies reacted perfectly with MSP-9 VLPs and the positive control *P. berghei* antigens, whereas no reaction was observed from the negative control TS antigen, indicating that the MSP-9 protein in VLPs was properly folded.

### 3.2. P. berghei-Specific IgG Antibody Response

To evaluate the IgG antibody levels in sera, mice blood was collected 4 weeks after prime and boost immunization. As shown in Figure 2A, mice immunized with MSP-9 VLPs showed higher levels of *P. berghei*-specific IgG antibody response after immunization compared to non-immunized mice. Sera from mice immunized with MSP-9 VLPs after prime and boost also reacted to MSP-9 VLPs (Figure 2B) and M1 VLPs (Figure 2C).

### 3.3. T Cell Responses in Blood

To determine T cell (CD4^+^ and CD8^+^) responses in immunized mice before and after the challenge infection, FACS analysis was performed using mouse blood (Figure 3A). As shown in Figure 3B,C, higher populations of CD4^+^ and CD8^+^ T cells were found in MSP-9 VLP-immunized mice after prime and boost (* *p* < 0.05, ** *p* < 0.01) compared to naïve control mice. Interestingly, high levels of CD4^+^ T cell populations were observed in the MSP-9 VLP-immunized group after the challenge infection (Figure 3B, * *p* < 0.05). These results indicate that MSP-9 VLP immunization induced CD4^+^ and CD8^+^ T cells and maintained high levels of CD4^+^ T cell populations after the challenge infection.

### 3.4. Memory T and B Cell Responses

To investigate the memory T and B cell responses in the spleen, mice were sacrificed at day 6 after *P. berghei* challenge infection. As seen in Figure 4, higher populations of CD4^+^ and CD8^+^ effector memory T (T_EM_) cells were found in MSP-9 VLP (MSP-9 VLP-immunized mice) compared to naïve challenged mice (Figure 4A,C; * *p* < 0.05, ** *p* < 0.01). As shown in Figure 5, higher populations of memory B cells were also found in MSP-9 VLP compared to naïve challenged mice (Figure 5A, * *p* < 0.05). These results indicate that the MSP-9 VLP vaccine contributed to a significant increase in T_EM_ and memory B cells.

### 3.5. Pro-Inflammatory Responses and Parasitemia against P. berghei Challenge Infection

Spleen samples were collected at 6 days post challenge with *P. berghei* to determine the level of the inflammatory cytokines IFN-γ and IL-6. As shown in Figure 6A,B, higher levels of IFN-γ and IL-6 were detected in naïve challenged mice. However, MSP-9 VLP-immunized mice showed a significant reduction in IFN-γ and IL-6 levels compared to naïve challenged mice (** *p* < 0.01). Parasitemia levels in blood were measured from all groups, including naïve mice, naïve challenged mice, and MSP-9-immunized mice (Figure 6C). At day 6 after challenge infection, naïve challenged mice showed 7% parasite counts compared to MSP-9 VLP-immunized mice (1.5%; ** *p* < 0.01). A MSP-9 VLP vaccine significantly increased survival time (6 days) compared to non-immunized control (naïve challenge), in which mice died within 19 days after *P. berghei* infection (Figure 6D).

## 4. Discussion

Merozoite surface protein-9 (MSP-9) is found in parasitophorous vacuoles and is an acidic basic repeat antigen (ABRA) that plays an important role in the survival of parasites in host cells. The ABRA is located on the schizont and merozoite surfaces and is also found in merozoite clusters agglomerated with immune serum [5,6,33]. This suggests that MSP-9 is a protein that has high immunogenicity and is required for parasites in the host. Immunogenicity refers to the ability of an antigen to induce an immune response. In our present study, MSP-9 VLPs were recognized by *P. berghei* antibody by western blot, and MSP-9 VLP immunization elicited *P. berghei*-specific IgG antibody responses, resulting in significantly reduced parasitemia in the blood upon *P. berghei* challenge infection. These results indicated that MSP-9 VLPs are highly immunogenic. These results are consistent with the findings from our previous study, which demonstrated that vaccination with virus-like particles containing AMA-1 of *P. berghei* elicits parasite-specific humoral and cellular immunity and protection [23]. Since MSP-9 VLP immunization induced IgG antibody responses against *P. berghei* antigens, which encompasses the MSP-9 protein, this indicates that MSP-9-specific antibodies were involved in protective immunity. To test whether antibodies induced by *P. berghei* can recognize whether MSP-9 protein in the VLPs was properly folded, coated MSP-9 VLPs were reacted with *P. berghei* antibody using ELISA, as previously described [34]. As expected, *P. berghei* antibody recognized the MSP-9 VLPs, indicating that the epitope regions of MSP-9 protein in the VLPs were similar to the epitope of *P. berghei*.

The vaccine should not only effectively elicit antigen-specific antibody responses but also ensure memory T and B cell responses [35,36]. Antibodies are known to play important roles in controlling blood stage infections [37,38], and memory T and B cells rapidly build systems that neutralize pathogens at the same time as pathogen invasion [39,40,41,42]. In our current study, MSP-9 VLP vaccination elicited significantly higher levels of IgG antibody and CD4^+^ and CD8^+^ effector memory T (T_EM_) cell and memory B (MB) cell responses. T_EM_ cells have an immediate effector function and can quickly migrate to peripheral tissues to provide antigen removal [43], while MB is important for maintaining a humoral immune response [42]. Consistent with these findings, MSP-9 VLP vaccination induced parasite-specific IgG antibody responses and immunological memory responses, indicating that these immune responses contribute to the significant reduction in parasite burden in the blood (parasitemia) and prolong survival time.

Increased malaria parasite loads in the blood stage exacerbates the pro-inflammatory response associated with severe malaria [44,45,46]. In our previous study, *Toxoplasma gondii* VLP-immunized mice demonstrated significantly lessened IFN-γ and IL-6 production following *T. gondii* challenge infection compared to the unimmunized control group [47]. Consistent with this finding, in our current study, VLP-immunized mice showed significant reductions in pro-inflammatory cytokine IFN-gamma and IL-6 responses compared to the naïve challenge control, which corresponds with diminished parasite loads and prolonged survival time in vaccinated mice.

Malaria infection reveals the exhaustion of parasite-specific CD4^+^ and CD8^+^ T cells, which are mediated by the programmed cell death-1 (PD-1) pathway [38]. T cell exhaustion signifies the absence of sterile immunity against blood stage malaria during malaria infection [38]. In our current study, VLP vaccination significantly increased CD4^+^ and CD8^+^ T cell responses. High levels of CD4^+^ and CD8^+^ T cells formed prior to infection allow for effective warfare against *Plasmodium* infection. These results indicated that VLP vaccination provided protection by inducing high levels of CD4^+^ and CD8^+^ T cell responses.

VLP vaccines adjuvanted with MCT were shown to enhance vaccine efficacy against *Plasmodium berghei* expressing *P. vivax* TRAP [48]. Mice immunized with a VLP vaccine without MCT adjuvant showed no protection, indicating the importance of MCT in inducing vaccine efficacy. In our current study, mice immunized with VLPs alone showed immunogenicity and protection, thereby implying that VLP vaccines formulated with adjuvants could result in better protection.

## 5. Conclusions

In summary, we investigated the vaccine efficacy induced by a *P. berghei* MSP-9 VLP vaccine in mice. We found that MSP-9 VLP vaccination elicited higher levels of *P. berghei*-specific IgG antibody responses and CD4^+^ and CD8^+^ T cell (containing effector memory T cells) and memory B cell responses. MSP-9 VLP vaccination also reduced pro-inflammatory cytokine IFN-γ and IL-6 levels and parasitemia, resulting in increased survival time. These results indicated that virus-like particle vaccines represent a promising strategy for the development of an effective vaccine to control the spread of malaria infection.

## Figures and Tables

**Figure 1 vaccines-08-00428-f001:**
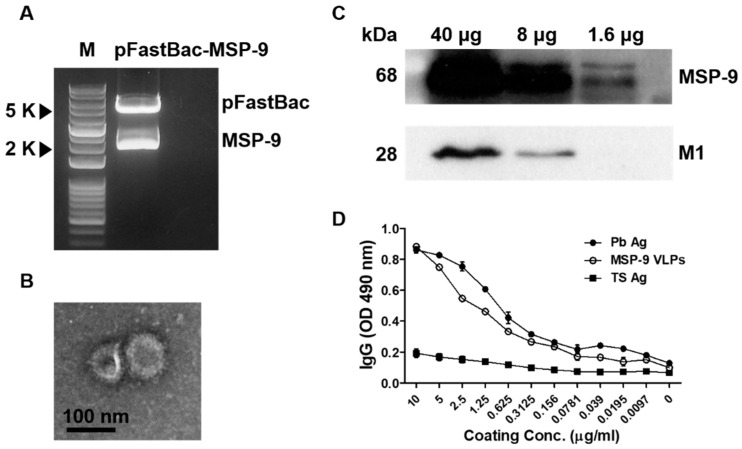
Recombinant plasmid pFastBac-MSP-9 digestion and characterization of MSP-9 VLPs. The pFastBac-MSP-9 plasmid was obtained by cloning the *P. berghei* optimized MSP-9 gene into a pFastBac vector with *EcoR*I/*Hind*III enzymes (**A**; M: DNA marker; size of vector: 4775 bp; size of MSP-9: 1857 bp). VLP images (**B**). VLPs (1.6, 8, 40 µg) were loaded for SDS-PAGE and western blot (**C**). Polyclonal mouse anti-*P. berghei* antibody and anti-M1 monoclonal antibody were used to probe *P. berghei* MSP-9 and influenza M1 protein, respectively. Serially diluted MSP-9 VLPs were reacted with *P. berghei* antibodies, in which *P. berghei* was used as the positive control and TS antigens as the negative control (**D**).

**Figure 2 vaccines-08-00428-f002:**
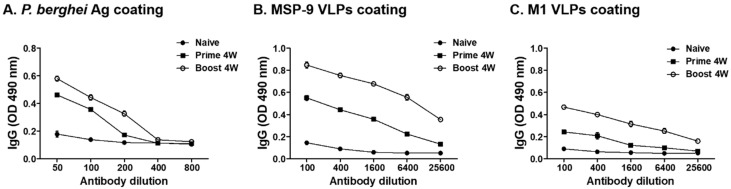
IgG antibody response upon intramuscular immunization with MSP-9 VLPs. Mice sera were collected twice at a 4-week interval after prime and boost immunization with MSP-9 VLPs. A significant increase in the *P. berghei*-specific IgG antibody level was found in MSP-9 VLP-immunized mice after prime and boost compared to non-immunized mice (**A**). MSP-9 VLP-immunized sera also reacted highly to MSP9-VLPs (**B**) and M1 VLPs (**C**).

**Figure 3 vaccines-08-00428-f003:**
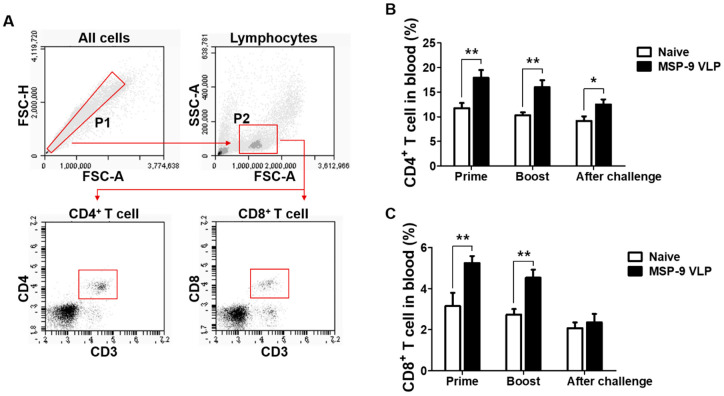
CD4^+^ and CD8^+^ T cell responses in blood. Gating strategy for CD4^+^ and CD8^+^ T cell responses (**A**). P1 is gating single cells and P2 is lymphocytes in blood (5000 cells). Populations of CD4^+^ (**B**) and CD8^+^ (**C**) T cells were analyzed by flow cytometry at prime and boost after immunization and 6 days after challenge with *P. berghei*. Higher populations of CD4^+^ and CD8^+^ were detected in MSP-9 VLP-immunized mice compared to naïve mice (* *p* < 0.05, ** *p* < 0.01). The error bars represent SEM.

**Figure 4 vaccines-08-00428-f004:**
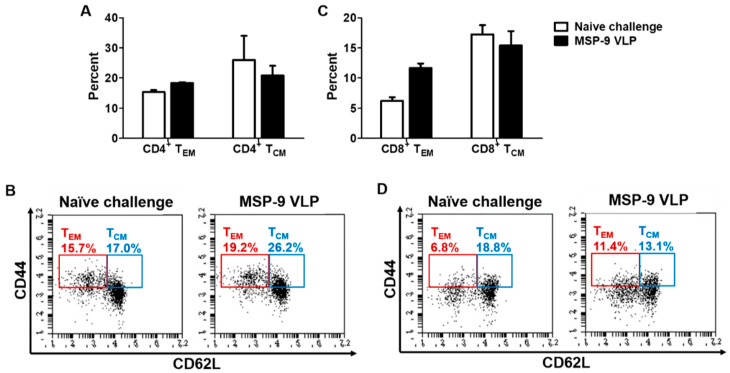
Memory T cell responses in spleen. Populations of CD4^+^ and CD8^+^ memory T cells were analyzed at day 6 post challenge using flow cytometry. MSP-9 VLP showed higher CD4^+^ (**A**) and CD8^+^ (**C**) effector memory T cell responses compared to the naïve challenged group. Numbers indicate the percentage of cell populations in each quadrant (**B**,**D**). The error bars represent SEM.

**Figure 5 vaccines-08-00428-f005:**
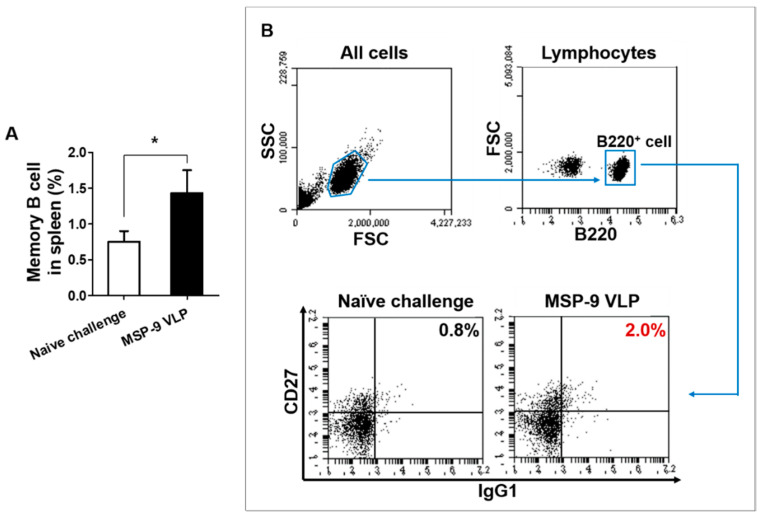
Memory B cell responses in spleen. Memory B cells were separated by CD45R/B220^+^, CD27^+^, and IgG1^+^ staining. A significantly higher level of memory B cells was found in MSP-9 VLP-immunized mice (MSP-9 VLP) compared to naïve challenged mice (naïve challenge) (**A**, * *p* < 0.05). Flow cytometry plots showing the gating strategy to identify memory B cells (**B**). The error bars represent SEM.

**Figure 6 vaccines-08-00428-f006:**
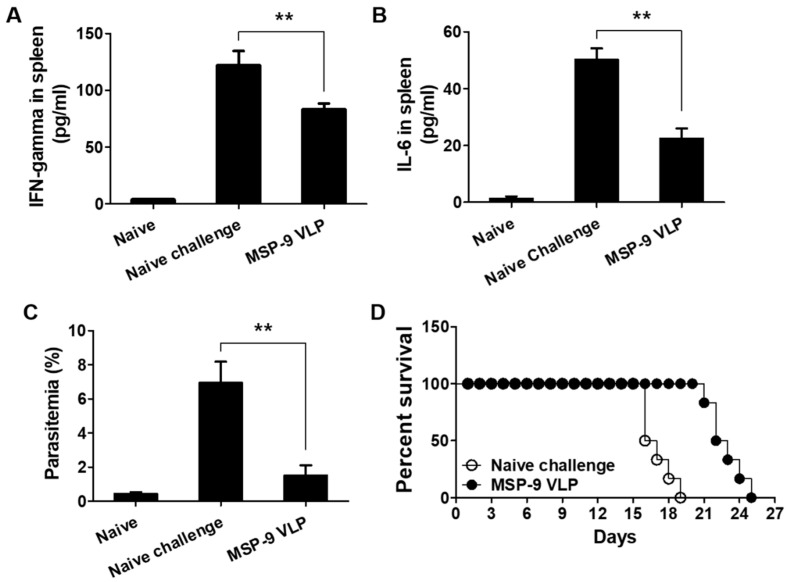
Pro-inflammatory response, parasitemia, and survival. To determine the level of the inflammatory cytokines IFN-γ (**A**) and IL-6 (**B**), mouse spleen was collected at day 6 after challenge with *P. berghei* (ANKA). Higher levels of IFN-γ and IL-6 were detected in spleen from naïve challenged mice (naïve challenge) compared to MSP-9 VLP-immunized mice (MSP-9 VLP, ** *p* < 0.01). Mice immunized intramuscularly (IM) with the MSP-9 VLP vaccine were challenged intraperitoneally (IP) with *P. berghei* (ANKA). At 6 days after *P. berghei* infection, parasites in red blood cells were collected and counted by flow cytometry (**C**, ** *p* < 0.01). Survival rates of mice were determined daily after *P. berghei* challenge infection (**D**). The error bars represent SEM.

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
