# Peer review of "Protective Immunity Induced by Virus-Like Particle Containing Merozoite Surface Protein 9 of Plasmodium berghei"

_vaccines, 2020, doi:10.3390/vaccines8030428_

Round 1

Reviewer 1 Report

The ms under review, authored by Lee et al. is an effort to target Plasmodium's MSP-9 protein using a vaccine designed using a virus-like particle technique in Mice. The ms is very crisply written and has very good readability. The authors have tried to be as concise as possible throughout the ms which is usually lacking in similar ms. Although, Introduction can be elaborated a bit to explain the VLP vaccine design and the pathogenicity of P.berghei.

Some constructive recommendations are provided below that can help enhance the significance of the work.

  1. The methods merely mention the Transmission microscope used. However, in figure 1B, the image seems to be quite sharp. Was that direct observation or noise reduction/refinement tools like MIPAR/J-Image were used? If yes, please add them in the methodology along with the params used.
  2. Discussions:
    2.1 Most of the results are plain reporting and not a research perspective especially in Section 3.3, 3.4. Kindly add sentences to the results that can cite reasons for such success of the VLP vaccine in teh experiments.
    For eg. The discussion is missing on the IFN- γ and IL-6 response. A slight biological relevance (backed by literature) in light of cytokine response can be very helpful for the ms.

    2.2 Mice is used as a model organism for Humans. Therefore, it would be justified for the authors to extrapolate their results onto Human malarial pathogens (P.falciparum/P.vivax). Indeed the target may be different in Humans but discussions are very shallow in their current form.
    Such information can be very useful for authors to suggest the future direction of their results. Moreover, they can also compare their results with RTS,S (P.falciparum vaccine using VLP) for eg. with the work in PMCID: PMC5492007
  3. Line 280 TEM- The EM should be subscripted to differentiate it from Transmission Electron Microscopy.

Reviewer 2 Report

The article, "Protective Immunity Induced by Virus-Like Particle Containing Merozoite Surface Protein 9 of Plasmodium berghei", is a significant research development of a vaccine against MSP-9 protein of Plasmodium. 

The reviewer would like the authors to address the following: 

  • It is not clear how a decrease in pro-inflammatory cytokines IFN-gamma and IL-6 will prolong survival time? These cytokines are known to argument antigen presentation and immune response. 
  • Antibody recognition of MSP-9 protein in ELISA does not confirm the proper folding of the entire protein but just a region. This needs to be modified. 
  • Western blot (Fig. 1C) shows M1 protein expression at least 10X less than MSP-9. The antibody dilution scale is the same for both (Fig. 2B and C). Does this reflect any difference in the immunogenicity of these two proteins? Please discuss. 
  • The rationale for intraperitoneal (antigen) and intramuscular (infection)   route needs to be added. 
  • The information on how much blood was collected, cell isolation technique,  number of cells sorted for CD4, and CD8 FACS analyses are missing.
  • Page 2, line 57, what's the source of AMA-1 protein? 
  • Page 3, line 130, P. berghei is misspelled. 
  • Page 5, Line 191 MSP-9 is misspelled. 

Reviewer 3 Report

The review of the manuscript “Protective Immunity Induced by Virus-Like Particle Containing Merozoite Surface Protein 9 of Plasmodium berghei” by Su-Hwa Lee, Hae-Ji Kang, Ki-Back Chu, Swarnendu Basak, Dong-Hun Lee, Eun-Kyung Moon and Fu-Shi Quan

The authors are focused on the merozoite surface protein because it plays an important role in the invasion of red blood cells, a process directly related to the occurrence of malaria disease. The authors note that the “merozoite surface protein 9 (MSP-9) is one of the antigens located within the merozoite surface and parasitophorous vacuole of the infected erythrocytes.”  The authors also cite “experimental evidence that MSP-9 participates in merozoites release from infected erythrocytes and binds to the human erythrocytes surface via N-terminal cysteine-rich region or high binding activity peptides (HBAPs).” The authors note that “MSP-9 has been suggested to be an important protein in vaccine development due to its role in erythrocyte invasion and immunogenicity in animal models”.

In their current research, the authors are focused on the development of malaria vaccine via the virus-like-particles (VLPs) because such particles induce an immune response, without the danger of viral infection. For that aim, as they explained in the introduction, they used merozoite surface protein 9. As they are still in the initial phase of the vaccine development, they use BALB/c mise as the animal model, and instead of MSP-9 from the human parasite Plasmodium falciparum, they use MSP-9 from the murine malaria Plasmodium berghei. They created VLPs with MSP-9 on the surface and introduced such particles in the model animals (intramuscular immunization).

The results reported clearly show that their approach induces IgG immune response higher than in non-immunized mice and that CD4+ and CD8+ T cells and effector memory cells have a higher concentration than in the non-immunized mice. The authors’ conclusion is that “the MSP-9 VLPs vaccine contributed a significant increase of TEM and memory B cells.”

The authors also showed that “MSP-9 VLPs immunized mice showed a significant reduction in IFN- γ and IL-6 levels compared to naïve challenged mice.”

The authors results are quite convincing, but the presentation lacks a quantitative comparison to other malaria vaccine studies based on the mice animal model and the murine malaria Plasmodium berghei. I would like to see a sentence or two, or better a paragraph addressing this issue.

Also, the authors claim that the MSP-9 protein in VLPs was properly folded, and used ELISA test to prove that. They exposed VLPs to P. berghai antibody and confirmed that antibodies recognize MSP-9. This, for me, is not enough to prove that protein has taken the same fold as in the viral particle. VLP particles probably have different shape and composition that the real parasite particle. The difference between parasite particle and VLP can induce a difference in the lateral strain to the MSP-9 protein that can affect its fold, as well as the protein’s dynamics, which can have an adverse effect on the binding affinity. I would ask the authors to address this, not experimentally but through discussion.

Minor comment. Lines 102-103. The sentence “For influenza M1 gene cloning, procedures were described previously (reference).” lacks reference.

I can suggest this manuscript for publication after the issues I have raised have been addressed.
